# Spatial Fuzzy C-Means Clustering Analysis of U.S. Presidential Election and COVID-19 Related Factors in the Rustbelt States in 2020

Shianghau Wu

Department of International Business, Chung Yuan Christian University, Taoyuan City 320314, Taiwan; antonwoo888@hotmail.com

**Abstract:** The rustbelt states play a key role in determining the vote turnout in the U.S. elections. The current study attempts to utilize the spatial fuzzy C-means method to analyze the U.S. presidential election in the rustbelt states in 2020. We intend to explore that the U.S. presidential election had related factors, including COVID-19-related factors, such as the mask-wearing percentage and the COVID-19 death tolls in each county of the rust belt states. Contrary to the related literature, the study uses education level, number of house units, unemployment rate, household income, COVID-19-related factors and the share of Republican's votes in the presidential election. The results indicate that spatial generalized fuzzy C-means analysis has better clustering results than the C-means clustering method. Moreover, the COVID-19 death toll in each county did not affect the Republican's vote share in the rustbelt states, while the mask-wearing behavior in some regions had a negative impact on the Republican's vote share.

**Keywords:** spatial fuzzy C-means; COVID-19; rustbelt states

**MSC:** 03B52; 03C45

## 1. Introduction

The U.S. presidential election in 2020 was influenced by the COVID-19 pandemic, including increasing infections, death tolls, and lockdowns. The previous literature indicated that political polarization was aggravated due to intense fear during the disaster [1,2]. People tended to search for assuage by insisting on their conservative political viewpoints and supporting the ruling party, while other scholars believed that some voters would punish the political elite for worse management during the natural or man-made disaster. Since COVID-19-related policies were created in a very short period of time, without full deliberation, it was possible to arouse public discontent [3]. People were more supportive of their governments during the early stage of the COVID-19 pandemic [4]. However, the evaluations of the policies about the pandemic were influenced by two polarized mindsets. Some voters chose to punish the politicians for the conditions caused by the pandemic, which were out of their control, while some voters were attentive to the political elites' reactions and determined their feelings accordingly [5].

The previous literature about the U.S. presidential election in 2020 focused on the effects of COVID-19 on the U.S. presidential election results. Hart (2021) stated that the COVID-19 pandemic seemed to have decreased the support for Trump among the Democrats, while it increased for independent voters [6]. Baccini et al. [7] pointed out that COVID-19-related factors negatively affected Donald Trump's re-election, and the effect was stronger in urban areas. They also observed that COVID-19 had a positive effect on the voters' mobilization for Joe Biden. The rustbelt states are traditionally "swing states" in the U.S. presidential elections, including Illinois, Wisconsin, Indiana, Michigan, Ohio, West Virginia, Pennsylvania, and New York. Geographical and racial divergences

increased in the counties of rustbelt states in the past five years [8]. The geographical factors enable these divergences to become more visible, and people tend to live in more politically polarized conditions [9]. The voting results of rustbelt states have a pivotal influence on the whole country. However, there are fewer instances in the literature about the voting results' analysis of the rustbelt states. Gimpel [10] pointed out that some counties in rustbelt states changed their support to the Democrats in the presential election in 2020. The influencing factors of the voting results need to be examined. In order to analyze the topic more thoroughly, we attempt to analyze the COVID-19 pandemic effects along with the regional factors' influence, the related economic variables, and the Republican's support rate in the 2020 U.S. presidential election.

The structure of this research is as follows: the Research Method Section presents our research design and related descriptive statistics of the variables. The Discussion Section presents the results of the research model. The research findings are listed in the Conclusions Section.

## 2. Methodology

### 2.1. Research Method

The current study used the spatial fuzzy C-means clustering method to analyze the influencing factors of COVID-19 on the U.S. presidential election. In order to explore the impacts of COVID-19 and other factors, such as social and geographical factors, as the mentioned in the Introduction, the study also used educational level, number of house units, unemployment rate, and household income variables to create the clustering. The previous literature utilized daily experience sampling (ESM) to analyze the impact of COVID-19 on employee uncertainty [11]. Di Nardo et al. (2019) utilized the literature review method to provide useful information about COVID-19 infection on neonates and children [12]. Regarding the fuzzy clustering approach, Indelicato et al. (2022) used the method with the fuzzy TOPSIS model to analyze the determinants of immigrants in Cuenca, Ecuador [13]. Compared to the COVID-19-related research about its effects on U.S. elections, the study considered spatial factors and attempted to describe the regional differences under the influence of these variables.

### 2.2. Data Description

The study explored the influencing factors of the pandemic on the 2020 U.S presidential election. The study used the Republican's voting share ($X_1$) in the U.S. presidential election in 2020 as one of the variables related to the U.S. presidential election. The data were obtained from the web repository (https://github.com/tonmcg/US_County_Level_Election_Results_08-20 (accessed on 6 August 2022)); it collected the 2020 election results at the county level, which were scraped from the results published by Fox News, Politico, and the New York Times.

In order to measure mask-wearing behavior in the rustbelt states ($X_2$), the study used the dataset collected by the survey firm, Dynata. Dynata surveyed 250 thousand respondents in the U.S. between 2 and 14 July 2020. The survey asked the respondents whether or not they wore face masks often in public. The responses included "always", "frequently", "sometimes", "rarely", and "never", according to the descending frequency.

The variables ($X_3$, $X_4$, $X_5$, $X_6$) were obtained from the dataset of the U.S. Census Bureau. These variables were released on a flow basis throughout each year.

The study also used the death toll ($X_7$) before the U.S. presidential election as a COVID-19-related variable. Other variables included education level and household economic condition. The descriptive statistics of all the variables are listed in Tables 1 and 2:

**Table 1.** All variables used for clustering.

| Variable | Meaning |
|---|---|
| $X_1$ | Republican's share of votes in U.S. presidential election |
| $X_2$ | The share of respondents who thought they wore face masks often |
| $X_3$ | The number of housing units |
| $X_4$ | The number of residents who were high-school graduates or above |
| $X_5$ | Unemployment rate |
| $X_6$ | Household income |
| $X_7$ | Death toll of COVID-19 cases |

**Table 2.** Descriptive statistics of all variables.

| Statistic | N | Mean | St.Dev | Min. | Max. |
|---|---|---|---|---|---|
| $X_1$ | 669 | 0.662 | 0.127 | 0.120 | 0.900 |
| $X_2$ | 669 | 0.536 | 0.139 | 0.190 | 0.880 |
| $X_3$ | 669 | 52,630.64 | 135,268.2 | 1107 | 2,204,019 |
| $X_4$ | 669 | 34,032.23 | 84,810.53 | 616 | 1,314,995 |
| $X_5$ | 669 | 4.591 | 1.273 | 2.400 | 13.00 |
| $X_6$ | 669 | 52,867.07 | 12,235.31 | 26,278 | 115,301 |
| $X_7$ | 669 | 71.175 | 306.17 | 0 | 5517 |

*2.3. C-Means Clustering*

Initially, the study used the classical C-means method to create the fuzzy unsupervised classification. The fuzziness degree (m) was set at 1.5 in order to obtain the satisfied results. The classical C-means method includes the following two equations. The first equation is the updated values of membership in each iteration of $u_{ik}$ [14]:

$$u_{ik} = \frac{(||x_k - v_i||^2)^{\frac{-1}{m-1}}}{\sum_{j=1}^{c} (||x_k - v_j||^2)^{\frac{-1}{m-1}}} \qquad (1)$$

The center of the cluster is as follows:

$$v_i = \frac{\sum_{k=1}^{N} u_{ik}^m (x_k)}{\sum_{k=1}^{N} u_{ik}^m} \qquad (2)$$

In Equations (1) and (2), $x_k$ represents the observation of k's value, $v_i$ is the value of the center of the cluster i, c is the cluster number, and m is the index of fuzziness.

*2.4. Fuzzy C-Means Clustering*

Fuzzy C-means clustering is an algorithm that permits a data point to pertain to two or more clusters. Let $X = \{x_1, x_2, \ldots, x_n\}$ represent an image with n pixels, where $x_i$ is the gray value of the ith pixel. The objective function of the standard FCM algorithm is as follows:

$$J = \sum_{k=1}^{K} \sum_{i=1}^{n} u_{ki}^m ||x_i - v_k||^2 \qquad (3)$$

In Equation (3), the center of the kth cluster is $v_k$ ($1 \leq k \leq K$), and $u_{ki}$ ($1 \leq k \leq K$, $1 \leq i \leq n$) is the membership degree function value of the ith pixel, which pertains to the kth cluster. $u_{ki}$ also needs to meet the requirements of the following constraints:

$$\sum_{k=1}^{K} u_{ki} = 1, \quad u_{ki} \in [0, 1], \quad 0 \leq \sum_{i=1}^{n} u_{ki} \leq n \qquad (4)$$

In Equation (3), the distance between $x_i$ and $v_k$ is used in the Euclidean form, and parameter m (m > 1) is a weighting parameter that relates to the level of fuzziness and the resulting partition. The minimization of the objective function in Equation (3) can obtain the updated equations of the membership degree function $u_{ki}$ and the cluster center $v_k$ as follows:

$$u_{ki} = \frac{1}{\sum_{i=1}^{k} \left( \frac{||x_i - v_k||^2}{||x_i - v_l||^2} \right)^{\frac{1}{m-1}}} \tag{5}$$

$$v_k = \frac{\sum_{i=1}^{n} u_{ki}^m x_i}{\sum_{i=1}^{n} u_{ki}^m} \tag{6}$$

The goal of these functions is to obtain suitable clusters for the data points.

*2.5. Spatial Fuzzy C-Means Clustering*

Fuzzy C-means clustering (FCM) has shortcomings due to its sensitivity to noise. Some algorithms were developed to overcome this shortcoming by utilizing the spatial information obtained from the neighborhood window around each pixel. Mean spatial information and median spatial information are two prevalent types of local information. The mean spatial information of the ith pixel is denoted as follows [15]:

$$\delta_i = \frac{1}{|S_i|} \sum_{p \epsilon S_i} x_p \tag{7}$$

In Equation (7), $S_i$ is the set of neighboring pixels in a window centered at the ith pixel, and $|S_i|$ represents its cardinality. The median spatial information can be represented as:

$$\varepsilon_i = median \{x_p\}, \; p \epsilon S_i \tag{8}$$

Most of the FCM algorithms utilize the above-mentioned local spatial information in the objective function; however, FCM algorithms with local spatial information can obtain a better image segmentation performance with a low noise level. The local spatial information obtained from the near pixels of a pixel is not efficient due to possible contamination. In fact, there are many pixels with a similar neighborhood configuration in an image. It is more beneficial to utilize pixels with a similar neighborhood configurations to the given pixel to obtain the spatial information than only using the neighboring pixels of the given pixel. Such types of spatial information can be taken as non-local spatial information. The non-local spatial information for the ith pixel $\overline{x_i}$ is calculated by the following equation [16]:

$$\overline{x}_i = \sum_{j \in w_i^r} w_{ij} x_j \tag{9}$$

In Equation (9), $\omega_i^r$ represents the r × r search window centered at the ith pixel. The non-local spatial information of the ith pixel is computed by using the pixels in the window. The weight between the ith and jth pixels can be denoted as $w_{ij}$ $(j \in w_i^r)$, $0 \leq w_{ij} \leq 1$ and $\sum_{j \in w_i^r} w_{ij} = 1$. The weight $w_{ij}$ is defined as follows:

$$w_{ij} = \frac{1}{Z_i} exp(-||x(N_i) - x(N_j)||_{2,\sigma}^2 / h^2) \tag{10}$$

In Equation (10), h means the filtering degree parameter and directs the decreasing weight function $w_{ij}$, and $Z_i = \sum_{j \in w_i^r} exp(-||x(N_i) - x(N_j)||_{2,\sigma}^2 / h^2)$ is the normalizing constant. The weight $w_{ij}$ depends on the similarity between the ith and jth pixels. The similarity is computed by the Gaussian weighted Euclidean distance $||x(N_i) - x(N_j)||_{2,\sigma}^2$. The positive term σ is the Euclidean distance, which means the standard deviation of the Gaussian kernel. $x(N_i)$ is the gray level vector with an s × s square neighborhood $N_i$ centered at ith pixel.

Fuzzy clustering algorithm with spatial information uses the spatial information for individual pixels to determine the spatial constant term, and then obtains the spatial constraint to the objective function of FCM.

## 3. Results

### 3.1. Fuzzy C-Means and Generalized Fuzzy C-Means Clustering

The study used the classical K-means to determine the number of clusters. According to Figure 1, the four clusters can explain almost 40% of the original data variance.

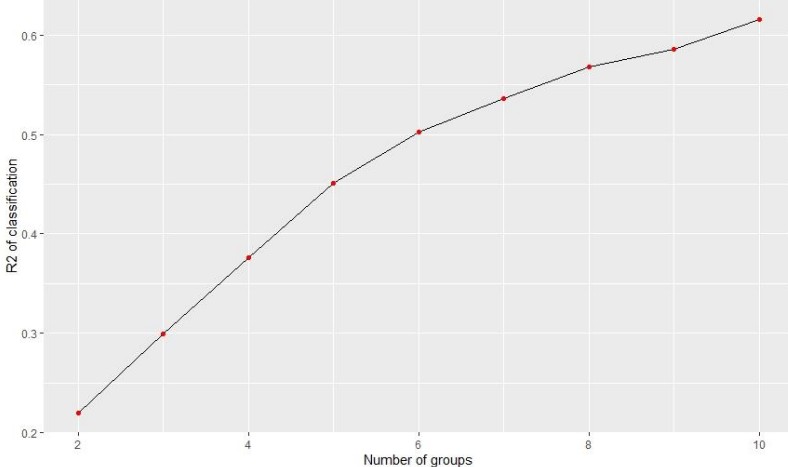

**Figure 1.** Impact of the number of groups on the explained variance.

Then, the study used the "fclust" package of R language to analyze the quality of the classification [17]. The study also utilized the "geocmeans" package of the R language to compute the generalized version of the c-means algorithm [18]. The algorithm can accelerate convergence and obtain less fuzzy results by adjusting the membership matrix at each iteration. It needs an extra beta parameter controlling the effectiveness of the modification. The modification only influences the formula updating the membership matrix.

$$u_{ki} = \frac{\left(||x_k - v_j||^2 - \beta_k\right)^{\frac{-1}{m-1}}}{\sum_{i=1}^{c}\left(||x_k - v_j||^2 - \beta_k\right)^{\frac{-1}{m-1}}} \tag{11}$$

In Equation (11), $\beta_k = min(||x_k - v||^2)$ and $0 \leq \beta \leq 1$. In order to choose an adequate value for this parameter, the study sought all the possible values between 0 and 1 with a step of 0.05. The results of the related index were obtained according to the ascending $\beta$ values in Table 3.

**Table 3.** Some indices with ascending $\beta$ values.

| Beta | Silhouette Index | Xie and Beni Index | Explained Inertia |
|------|------------------|--------------------|--------------------|
| 0 | 0.287 | 2.476 | 0.161 |
| 0.05 | 0.29 | 2.282 | 0.171 |
| 0.1 | 0.294 | 2.113 | 0.181 |
| 0.15 | 0.298 | 1.964 | 0.191 |
| 0.2 | 0.3 | 1.83 | 0.201 |
| 0.25 | 0.303 | 1.706 | 0.212 |
| 0.3 | 0.307 | 1.584 | 0.223 |
| 0.35 | 0.313 | 1.47 | 0.235 |
| 0.4 | 0.315 | 1.374 | 0.247 |
| 0.45 | 0.315 | 1.292 | 0.26 |
| 0.5 | 0.292 | 1.478 | 0.265 |

**Table 3.** *Cont.*

| Beta | Silhouette Index | Xie and Beni Index | Explained Inertia |
|------|------------------|--------------------|--------------------|
| 0.55 | 0.289 | 1.41 | 0.277 |
| 0.6 | 0.286 | 1.349 | 0.289 |
| 0.65 | 0.283 | 1.295 | 0.301 |
| 0.7 | 0.281 | 1.249 | 0.313 |
| 0.75 | 0.277 | 1.211 | 0.325 |
| 0.8 | 0.273 | 1.182 | 0.337 |
| 0.85 | 0.268 | 1.163 | 0.349 |
| 0.9 | 0.259 | 1.157 | 0.361 |
| 0.95 | 0.249 | 1.172 | 0.371 |
| 1 | 0.235 | 1.296 | 0.374 |

According to Table 1, the study chose beta = 0.8, maintained a satisfied silhouette index, increased the Xie and Beni index, and explained inertia. The results of GFCM (generalized version of fuzzy C-means clustering) and FCM are listed in Table 4.

**Table 4.** Comparison of the indices between GFCM and FCM.

| | GFCM | FCM |
|---|------|-----|
| Silhouette index | 0.273 | 0.287 |
| Partition entropy | 0.323 | 0.951 |
| Partition coeff | 0.837 | 0.486 |
| XieBeni index | 1.182 | 2.476 |
| Fukuyama Sugeno index | 1096.84 | 1706.23 |
| Explained inertia | 0.337 | 0.161 |

The results indicate that the GFCM provides a less fuzzy solution (with higher explained inertia and lower partition entropy), but keeps a good silhouette index and a lower Xie and Beni index. The study created two membership matrices maps and the most likely group for each observation. The study used the function map clusters from geocmeans in R language. We set a threshold of 0.45. If an observation only obtained values below this probability in a membership matrix, it was marked as "undecided" (represented by transparency on the map).

In Figure 2, the left-hand-side graph was the fuzzy C-means clustering result. The right-hand-side graph was the generalized fuzzy C-means clustering result. We can observe that the right-hand-side graph had fewer undecided parts.

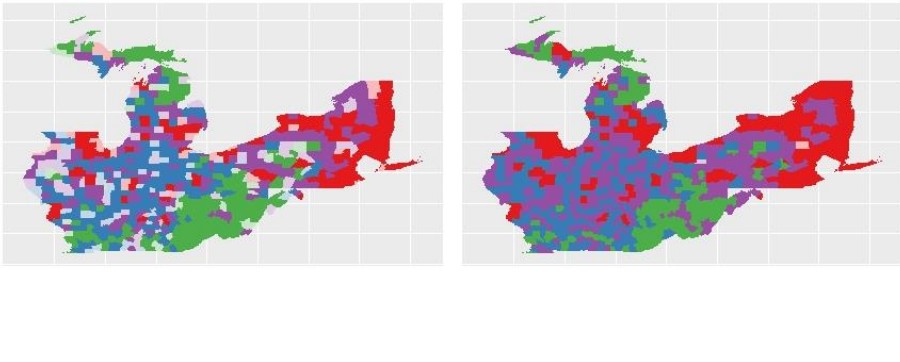

**Figure 2.** FCM and GFCM clusters.

### 3.2. Spatial C-Means and Generalized C-Means

The study used the SFCM function of R language to execute spatial c-means clustering. The first step was to determine a spatial weight matrix indicating the observations that were neighbors and the strength of their relationship. The study attempted to use a basic queen neighbor matrix (built with the spdep package of R language). The matrix should be row-standardized to ensure that the interpretation of all the parameters remains clear.

The two following equations indicate how the functions renewing the condition of the membership matrix and the centers of the clusters are modified.

$$u_{ik} = \frac{\left(||x_k - v_i||^2 + \alpha||\overline{x_k} - v_i||^2\right)^{\frac{-1}{m-1}}}{\sum_{j=1}^{c}\left(||x_k - v_i||^2 + \alpha||\overline{x_k} - v_i||^2\right)^{\frac{-1}{m-1}}} \tag{12}$$

$$v_i = \frac{\sum_{k=1}^{N} u_{ik}^m \left(x_k + \alpha\overline{x_k}\right)}{(1+\alpha)\sum_{k=1}^{N} u_{ik}^m} \tag{13}$$

In Equations (12) and (13), $\overline{x}$ is the lagged version of x, and $\alpha \geq 0$.

The SFCM (spatial fuzzy C-means) can be taken as a spatially smoothed version of the classical c-means, and alpha controls the degree of spatial smoothness. This smoothing can be taken as an attempt to reduce the spatial overfitting of the classical c-means.

The study chose the best alpha value in order to reduce spatial inconsistency as much as possible and to maintain a good classification quality. The relationship between the spatial inconsistency and alpha value is shown in Figure 3.

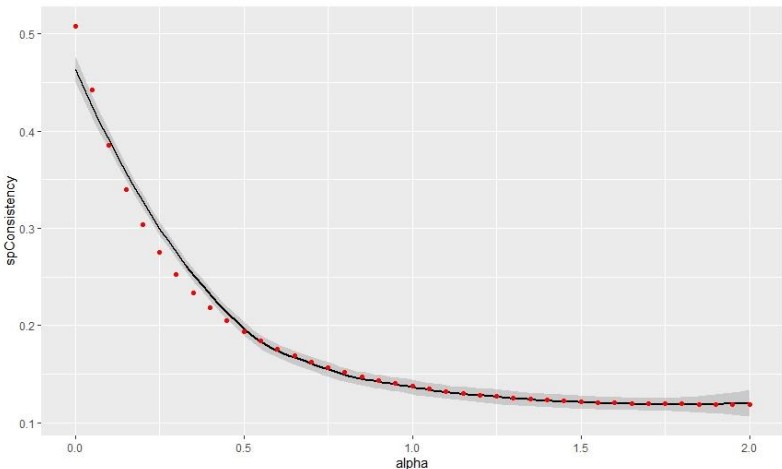

**Figure 3.** Link between alpha value and spatial inconsistency.

In Figure 3, the increasing alpha value results in the decrease in the spatial inconsistency.

In Figure 4, the explained inertia decreased when the alpha value increased and again followed an inverse function. The classification searched for a compromise between the original and lagged values. However, the loss was only 3% between alpha = 0 and alpha = 2.

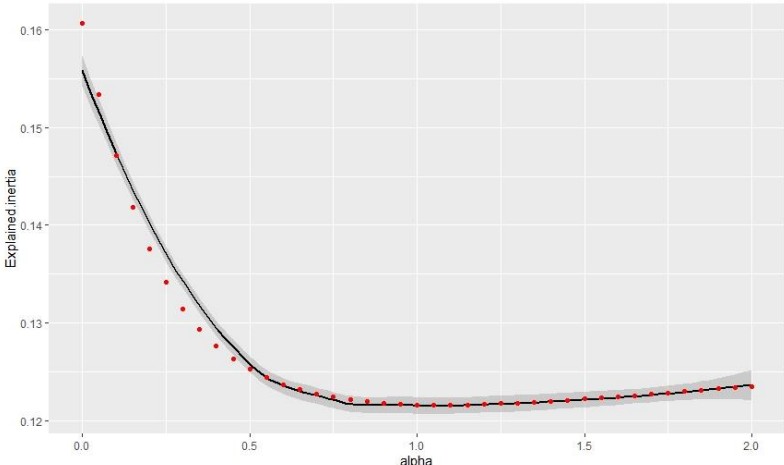

**Figure 4.** The relationship between the alpha value and explained inertia.

According to Figures 5 and 6, as a larger silhouette index means a better classification, and a smaller Xie and Beni index represents a better classification, the study intended to retain the alpha = 0.25 value to provide a good balance between spatial consistency and classification quality.

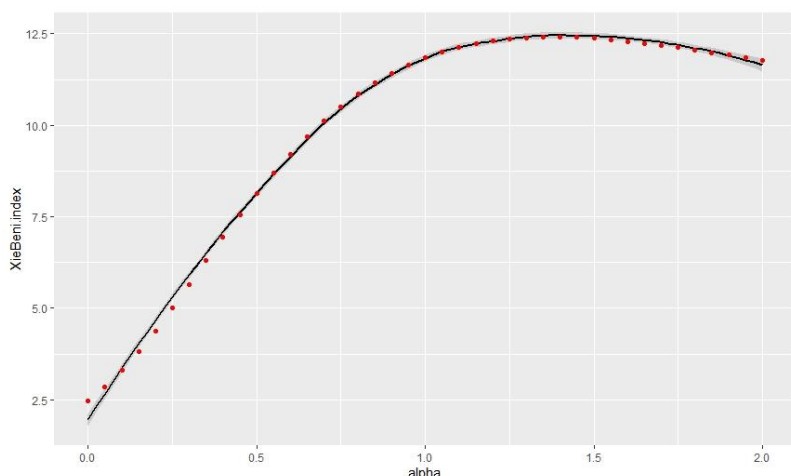

**Figure 5.** Link between alpha and Xie and Beni index.

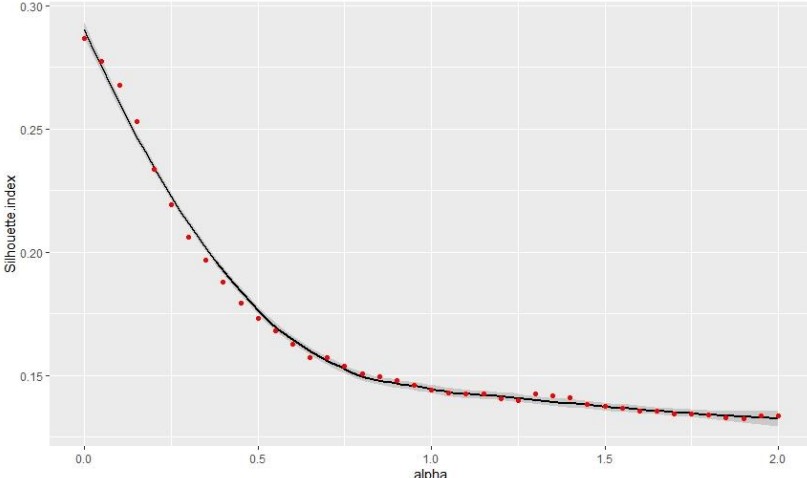

**Figure 6.** Link between alpha value and silhouette index.

### 3.3. Spatial Generalized Fuzzy C-Means (SGFCM)

In order to facilitate the clustering process of the SGFCM method, we needed to determine the alpha and beta values of the following equation regarding the center of the clusters.

$$u_{ik} = \frac{(||x_k - v_i||^2 - \beta_k + \alpha||\overline{x_k} - v_i||^2)^{\frac{-1}{m-1}}}{\sum_{j=1}^{c} (||x_k - v_i||^2 - \beta_k + \alpha||\overline{x_k} - v_i||^2)^{\frac{-1}{m-1}}} \tag{14}$$

The study attempted to use the multiprocessing approach to select the suitable alpha and beta values. The impact of alpha and beta values on the various indices is shown as follows:

Figures 7 and 8 indicate that some specific combinations of alpha and beta values generate good results in the range of 0.3 < alpha < 0.7 and 0.4 < beta < 0.6. Figure 9 shows that the selection of beta has no impact on spatial consistency.

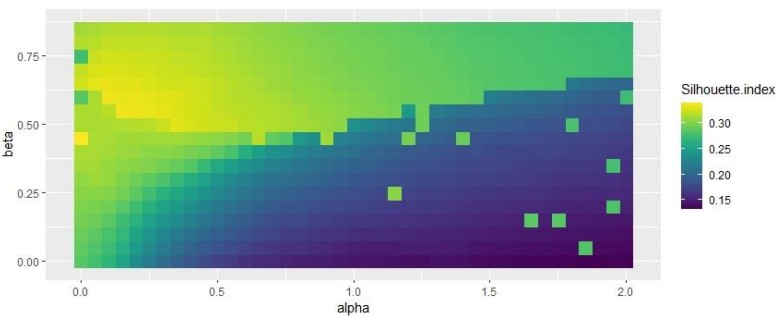

**Figure 7.** Influence of beta and alpha values on silhouette index.

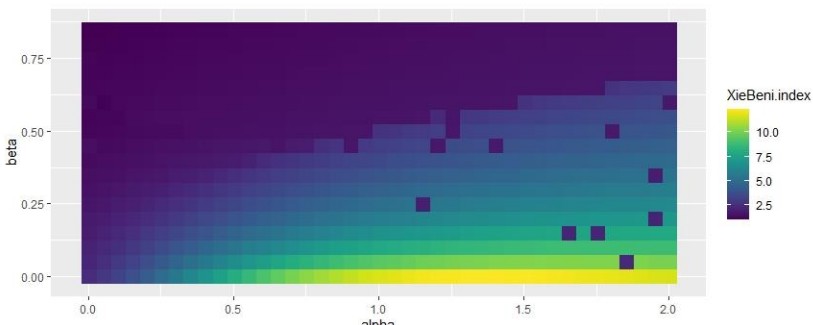

**Figure 8.** Influence of beta and alpha values on Xie and Beni index.

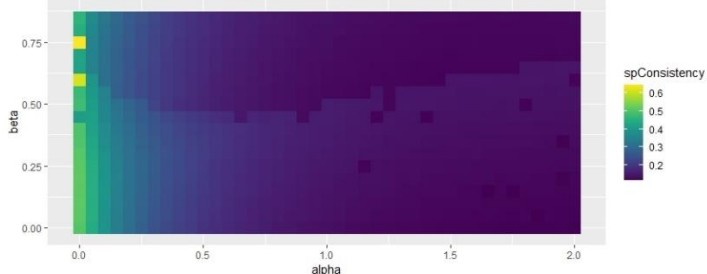

**Figure 9.** Influence of beta and alpha values on spatial inconsistency.

Regarding Figures 7–9, the study selected beta = 0.5 and alpha = 0.25, which obtained better results for all the indices considered. Based on the alpha and beta values, the study acquired the results of the SFCM and SGFCM results (see Table 5).

**Table 5.** Comparison of the indices between SFCM and SGFCM.

|  | SFCM | SGFCM |
|---|---|---|
| Silhouette index | 0.219 | 0.319 |
| Partition entropy | 1.043 | 0.682 |
| Partition coeff | 0.431 | 0.633 |
| XieBeni index | 5.008 | 1.394 |
| Fukuyama Sugeno index | 1824.58 | 1290.69 |
| Explained inertia | 0.134 | 0.248 |
| sp consistency | 0.276 | 0.262 |

The results of the SGFCM are better concerning the semantic and spatial aspects due to the lower partition entropy, Xie Beni index, and Fukuyama Sugeno index, and higher values of other indices.

The SFCM and SGFCM clustering maps are listed as follows.

According to Figure 10, the right-hand-side graph is the SGFCM clustering map. The left-hand-side graph is the SFCM clustering map. We can observe that the undecided units are less on the SGFCM clustering map.

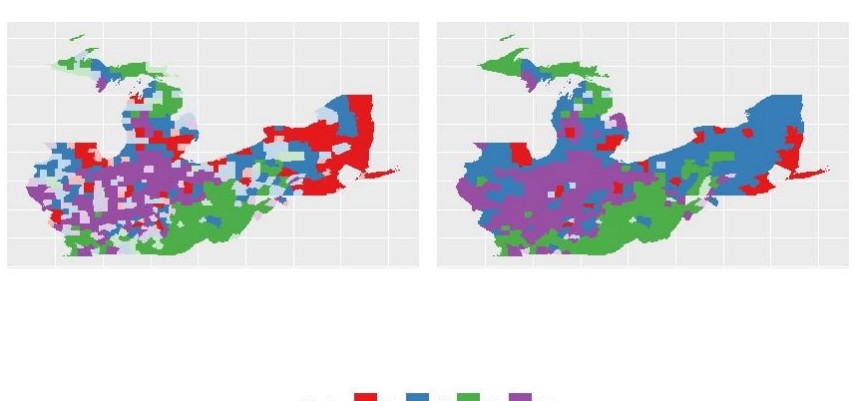

**Figure 10.** Most likely cluster and undecided units of SFCM and SGFCM.

*3.4. Comparison of the Four Algorithms*

The study attempted to perform a thorough spatial analysis and compare the spatial consistency of the four classifications (FCM, GFCM, SFCM, SGFCM) (see Table 6).

**Table 6.** Moran I index for the columns of the membership matrices among the four algorithms.

|  | FCM | GFCM | SFCM | SGFCM |
|---|---|---|---|---|
| Cluster 1 | 0.642 | 0.602 | 0.769 | 0.696 |
| Cluster 2 | 0.349 | 0.187 | 0.501 | 0.66 |
| Cluster 3 | 0.691 | 0.595 | 0.809 | 0.823 |
| Cluster 4 | 0.205 | 0.14 | 0.674 | 0.73 |

The Moran I value according to the membership matrices were higher for SFCM and SGFCM, representing strongaer spatial structures in the classifications.

The study also checked that the values of spatial inconsistency for SGFCM were significantly lower than those of SFCM. The study used the previously mentioned 250 values obtained by permutations; we could calculate a pseudo $p$-value = 0.032 > 1/250 = 0.004. This means that the SGFCM algorithm did not have a predominant advantage over the SFCM algorithm. However, the SGFCM clustering map indicated that the undecided points were fewer than that of the SFCM.

We can observe that the undecided parts were fewer as compared with Figures 2 and 10.

## 4. Discussion

The study attempted to utilize the spatial fuzzy C-means clustering method to analyze the relationship among COVID-19-related factors and the vote share of Republicans in the U.S. presidential election in the rustbelt states in 2020. The study found that spatial generalized fuzzy C-means clustering (SGFCM) produced better results compared to the other three algorithms according to Table 3. The study also found the SGFCM clustering graph in Figure 10 presented better results because the uncertain parts (areas that did not belong to any cluster) were fewer compared to the other clustering results shown in Figure 2.

The descriptive statistics of the four clusters (Tables A1–A4) are listed in the Appendix A. According to the four tables, we can conclude the four clusters are as follows:

(1)    First cluster: the cluster had lower $X_1$ (mean < 0.5), higher $X_2$, higher $X_4$, lower $X_5$, and higher $X_6$ values. Other variables did not seem obvious. We can conclude that people in this region were not inclined to support the Republican candidate, often wore masks, had more high-school graduates or above, had a lower unemployment rate, and a higher income. The first cluster included a little part of southeastern Pennsylvania, New York state and other scatter parts of the rustbelt states.

(2)    Second cluster: The cluster had higher $X_1$ (mean > 0.5), higher $X_2$, lower $X_4$, lower $X_5$, and higher $X_6$ values. Other variables did not seem obvious. We can conclude that people in this region were inclined to support the Republican candidate, often wore masks, had less high-school graduates, a lower unemployment rate, and higher income. The second cluster included the larger part of New York state, most part of Michigan and northern Illinois.

(3)    Third cluster: The cluster had higher $X_1$ (mean > 0.5), lower $X_2$, lower $X_4$, higher $X_5$, lower $X_6$, and higher $X_7$ values. This means that people in this region tended to support the Republican candidate, wore masks less frequently, had less high-school graduates or above, a higher unemployment rate, lower income, and higher COVID-19 death toll. The cluster included some parts of Kentucky, West Virginia and Ohio and other scatter parts of the rustbelt states.

(4)    Fourth cluster: The cluster had higher $X_1$ (mean > 0.5), lower $X_2$, lower $X_4$, lower $X_5$, higher $X_6$, and higher $X_7$ values. This means that people in this region tended to support the Republican candidate, wore masks less frequently, had less high-school graduates or above, a lower unemployment rate, higher income, and higher COVID-19 death toll. The cluster included the larger part of Indiana, Ohio and part of Illinois.

The results seem to slightly contrast with the previous literature. Warshaw et al. (2020) found that COVID-19 fatalities decreased the support for Donald Trump in the 2020 presidential election [19]. However, our results show that the third and fourth clusters in the rustbelt states have higher COVID-19 death tolls with higher Republican vote shares and residents less inclined to wear face masks. Meanwhile, the second cluster had higher Republican vote shares and the residents there often wore face masks, while the COVID-19 death toll seemed unimportant. We can conclude that the COVID-19 death toll in each county did not affect the Republican vote shares in the rustbelt states, while the mask-wearing behavior in some regions had a negative impact on the Republican vote shares.

According to Figure 11, we can observe that cluster 2 accounts for the largest area in the rustbelt states. Cluster 1 accounts for the smallest area. The clustering results indicate that the U.S. presidential election-related factors and COVID-19-related factors are closely related to the clustering results. It enables the researchers in the related field to conduct further studies.

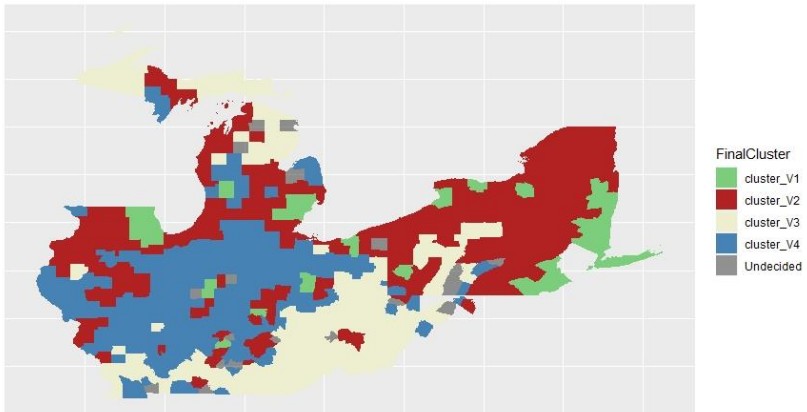

**Figure 11.** Final cluster of SGFCM.

## 5. Conclusions

The present study intended to use the spatial fuzzy C-means clustering to analyze the related factors of COVID-19 and the U.S. presidential election in the rustbelt states in 2020. The study found that the spatial generalized fuzzy C-means (SGFCM) method produced better clustering results. The SGFCM method divided the rustbelt states into four areas. The results imply that the COVID-19 death toll in each county did not affect the Republican vote shares in the rustbelt states, while the mask-wearing behavior in some regions had a negative impact on the Republican vote shares. It is worth conducting further research.

**Funding:** This research was supported by the Preliminary Research Resource Funding from Chung Yuan Christian University.

**Institutional Review Board Statement:** Not Applicable.

**Informed Consent Statement:** Not Applicable.

**Data Availability Statement:** The COVID-19-related data for the U.S. can be downloaded from https://github.com/nytimes/COVID-19-data (accessed on 6 August 2022). The U.S. presidential election results in each county can be downloaded from https://github.com/tonmcg/US_County_Level_Election_Results_08-20/blob/f9b5f335ad1c66a7eba681539db49eec0c22787b/2020_US_County_Level_Presidential_Results.csv (accessed on 6 August 2022). The education level and household economic condition can be downloaded from https://www.census.gov/ (accessed on 6 August 2022).

**Acknowledgments:** The author would like to show their gratitude for the preliminary research fund received by Chung Yuan Christian University.

**Conflicts of Interest:** The author declares no conflict of interest.

## Appendix A

**Table A1.** Descriptive statistics for cluster 1.

|  | $X_1$ | $X_2$ | $X_3$ | $X_4$ | $X_5$ | $X_6$ | $X_7$ |
|---|---|---|---|---|---|---|---|
| Q5 | 0.222 | 0.514 | 28 | 9872.6 | 2.9 | 46,288.2 | 7 |
| Q10 | 0.27 | 0.549 | 67 | 16,480.8 | 3.2 | 49,515 | 19 |
| Q25 | 0.37 | 0.641 | 198 | 41,764 | 3.4 | 58,222 | 45 |
| Q50 | 0.446 | 0.742 | 379 | 132,127 | 3.8 | 66,270 | 81 |
| Q75 | 0.533 | 0.788 | 501 | 211,597 | 4.2 | 86,108 | 103 |
| Q90 | 0.614 | 0.82 | 596 | 347,971.4 | 4.9 | 94,521 | 153 |
| Q95 | 0.678 | 0.842 | 632 | 522,061 | 5.4 | 100,887 | 165 |
| Mean | 0.448 | 0.71 | 342.689 | 168,655.4 | 3.901 | 70,973.15 | 80.35 |
| Std | 0.134 | 0.107 | 187.555 | 199,205.9 | 0.793 | 17,662.76 | 48.11 |

**Table A2.** Descriptive statistics for cluster 2.

|  | $X_1$ | $X_2$ | $X_3$ | $X_4$ | $X_5$ | $X_6$ | $X_7$ |
|---|---|---|---|---|---|---|---|
| Q5 | 0.417 | 0.449 | 49.4 | 4579.6 | 3.1 | 43,118 | 9 |
| Q10 | 0.463 | 0.487 | 89 | 5836.6 | 3.3 | 46,262 | 15 |
| Q25 | 0.539 | 0.54 | 198 | 11,116 | 3.8 | 49,767 | 37 |
| Q50 | 0.605 | 0.612 | 368 | 20,204 | 4.4 | 53,901 | 77 |
| Q75 | 0.674 | 0.723 | 510 | 41,229 | 4.9 | 60,121 | 115 |
| Q90 | 0.729 | 0.79 | 608 | 68,550 | 5.5 | 66,521 | 155 |
| Q95 | 0.762 | 0.827 | 633.6 | 109,462 | 5.7 | 73,006.8 | 174.2 |
| Mean | 0.599 | 0.627 | 356.193 | 35,019.47 | 4.415 | 55,596.35 | 79.845 |
| Std | 0.105 | 0.119 | 186.098 | 62,713.13 | 0.899 | 9592.194 | 52.128 |

**Table A3.** Descriptive statistics for cluster 3.

|  | $X_1$ | $X_2$ | $X_3$ | $X_4$ | $X_5$ | $X_6$ | $X_7$ |
|---|---|---|---|---|---|---|---|
| Q5 | 0.576 | 0.341 | 29.4 | 2415 | 3.84 | 30,950 | 7 |
| Q10 | 0.624 | 0.368 | 56 | 3297 | 4.2 | 33,218 | 13 |
| Q25 | 0.693 | 0.409 | 155 | 5072 | 4.9 | 38,171 | 43 |
| Q50 | 0.747 | 0.475 | 341 | 8354 | 5.6 | 43,457 | 81 |
| Q75 | 0.787 | 0.54 | 518 | 13,670 | 6.4 | 48,182 | 129 |
| Q90 | 0.83 | 0.611 | 604 | 25,221 | 7.4 | 51,812.2 | 169 |
| Q95 | 0.856 | 0.641 | 631.6 | 34,390.8 | 8.3 | 55,443.8 | 195 |
| Mean | 0.734 | 0.481 | 334.757 | 14,615.75 | 5.743 | 43,457.84 | 88.095 |
| Std | 0.088 | 0.097 | 199.112 | 47,465.07 | 1.377 | 8146.738 | 57.705 |

**Table A4.** Descriptive statistics for cluster 4.

|  | $X_1$ | $X_2$ | $X_3$ | $X_4$ | $X_5$ | $X_6$ | $X_7$ |
|---|---|---|---|---|---|---|---|
| Q5 | 0.555 | 0.285 | 35 | 3184.2 | 2.7 | 41,799.2 | 9 |
| Q10 | 0.602 | 0.33 | 60 | 4188.8 | 2.98 | 44,913 | 21 |
| Q25 | 0.673 | 0.392 | 146 | 6912 | 3.3 | 48,342 | 49 |
| Q50 | 0.728 | 0.462 | 289 | 11,761 | 4 | 52,798 | 97 |
| Q75 | 0.76 | 0.529 | 473 | 18,689 | 4.5 | 57,705 | 145 |
| Q90 | 0.789 | 0.584 | 585 | 33,791.4 | 5.1 | 63,827.4 | 175.4 |
| Q95 | 0.809 | 0.627 | 629.6 | 45,496 | 5.46 | 67,758 | 193 |
| Mean | 0.707 | 0.459 | 308.856 | 18,494.64 | 4.009 | 53,761.27 | 98.385 |
| Std | 0.084 | 0.106 | 190.401 | 46,181.82 | 0.91 | 8948.192 | 58.712 |

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
