# Peer review of "Spatial Fuzzy C-Means Clustering Analysis of U.S. Presidential Election and COVID-19 Related Factors in the Rustbelt States in 2020"

_axioms, doi:10.3390/axioms11080401_

Round 1
Reviewer 1 Report
This work aims to study the effects of COVID-19 on the 2020 US presidential election.
This topic is very interesting. It deals with a very dynamic issue between the various countries and provides a clear vision of the management of the pandemic.
The introduction is well-drafted on a theoretical level and positions the topic well in the context studied. It is also true that the section has gaps, as there is no mention of any methodology. I would advise the authors to add lines that mention the methodology, citing other similar works in other fields, such as Indelicato and Martin (2022), Di Nardo (2019), and Chawla (2021).
The data and methodology used are sparsely detailed in this document. The illustration of the data presents a lack of details. I do not understand how the variables x1, x3-x6 are measured. The methodology was introduced without any theoretical premise. Formulas appear attractive, but it is also essential to understand what they are for, especially (1) - (6).
The results are not commented on. The figures and tables are nice, but an appropriate description must follow them. This section is limited to illustrating graphs and tables. I would advise the authors to comment on each graph and table presented in this paper, to provide a more detailed view of the great work performed.
Discussions are not present in this study. Usually, the discussion section serves to offer a concise overview of the results by comparing them with other studies.
Minor comments
Line 34: Hart (?)
Author Response
- The author is gratitude for the reviewer 1’s precious comments. The author cites the literature of Indelicato and Martin (2022), Di Nardo (2019) and Chawla(2021) to mention the methodology.
- The author revised the methodology part and pointed out the difference of this research from other ones.
- The author added the definition of all variables and explained the meaning of eq.(1) to eq.(6).
- The results and discussion part were elaborated.
Reviewer 2 Report
In the present paper, the author analyzed the U.S. presidential election in the rustbelt states in 2020, via the spatial fuzzy C- Means clustering method. The proposed factors were based on COVID-19-related factors. The analysis of the clustering results indicated that there exists a relation between COVID-19 and the U.S. presidential election factors. As a result, the paper is well structured and provides interesting results. I recommend to be accepted after minor revisions.
1) As the author stated, the spatial generalized fuzzy C-Means clustering had better results than three other algorithms. Could the author present in the discussion section a more informative representation of this conducted comparison?
Author Response
The author would like to show the great gratitude for the views of reviewer 2. The author offered the response according to the comparison of the clustering results graph (Figure 2 and Figure 10). The study found that the clustering results graph of SGFCM clustering graph in Figure 10 had better results because the uncertain parts (areas which weren’t belong to any cluster) were fewer as compared with other clustering results shown in Figure 2.
Round 2
Reviewer 1 Report
The author has considered all comments. He adapted the paper according to the suggestions. The quality of the paper is now remarkable.